# The Transmission, Infection Prevention, and Control during the COVID-19 Pandemic in China: A Retrospective Study

**DOI:** 10.3390/ijerph19053074

**Published:** 2022-03-05

**Authors:** Lifeng Zhang, Roy E. Welsch, Zhi Cao

**Affiliations:** 1State Key Laboratory of Networking and Switching Technology, Beijing University of the Posts and Telecommunications, Beijing 100876, China; 2Sloan School of Management, Massachusetts Institute of Technology, Cambridge, MA 02139, USA; rwelsch@mit.edu; 3Center for Statistics and Data Science, Massachusetts Institute of Technology, Cambridge, MA 02139, USA; 4Computer Science Department, University of Massachusetts Boston, Boston, MA 02125, USA; zhi.cao001@umb.edu

**Keywords:** COVID-19, pandemic, transmission, infection prevention and control

## Abstract

The first wave of COVID-19 in China began in December 2019. The outbreak was quickly and effectively controlled through strict infection prevention and control with multipronged measures. By the end of March 2020, the outbreak had basically ended. Therefore, there are relatively complete and effective infection prevention and control (IPC) processes in China to curb virus transmission. Furthermore, there were two large-scale updates for the daily reports by the National Health Commission of the People’s Republic of China in the early stage of the pandemic. We retrospectively studied the transmission characteristics and IPC of COVID-19 in China. Additionally, we analyzed and modeled the data in the two revisions. We found that most cases were limited to Hubei Province, especially in Wuhan, and the mortality rate was lower in non-Wuhan areas. We studied the two revisions and utilized the proposed transmission model to revise the daily confirmed cases at the beginning of the pandemic in Wuhan. Moreover, we estimated the cases and deaths for the same stage and analyzed the effect of IPC in China. The results show that strong and effective IPC with strict implementation was able to effectively and quickly control the pandemic.

## 1. Introduction

Coronavirus disease 2019 (COVID-19) is an infectious disease caused by severe acute respiratory syndrome coronavirus 2 (SARS-CoV-2) [1,2,3,4], which is highly infectious and can be spread by droplets, through the air, etc. [5,6]. The virus enters the human body by contacting mucosal cells and attacking the body [7,8], producing symptoms including fever, dry cough, and shortness of breath. However, some individuals deteriorate rapidly and develop acute respiratory distress syndrome [9]. Additionally, it can lead to respiratory organ failure and a series of complications, especially for older people with underlying diseases [10,11,12]. Moreover, some studies have shown that some infected people who are asymptomatic may also be infectious [13,14,15]. At present, several variants of the virus have appeared, including Alpha, Beta, and Omicron. The variants have some differences in their characteristics, such as viral load, duration of infectiousness, and rates of reinfection [16].

Infection prevention and control (IPC) is a scientific approach and a practical solution designed to prevent harm caused by infection in patients and health workers [17]. Generally speaking, for infectious diseases, in the absence of effective vaccines [18,19], IPC is the most effective way to break the transmission chain of the virus [15,20]. Specifically, medical or non-medical measures should be taken to identify infected people to prevent them from spreading the virus to healthy people and reduce further transmission. Before infected people infect healthy people, they need to be identified and prevented from spreading the virus by implementing preventative measures, including reasonable social distance and isolation [21,22,23].

The first wave of the pandemic in China began in December 2019. By the end of March 2020, the first wave of the local outbreak had basically ended. Therefore, there is a relatively complete and effective IPC process in China for managing a pandemic. The beginning of the pandemic coincided with the eve of the Chinese Lunar New Year, and this period is known to correspond to the largest annual human migration in the world [24]. In the weeks that followed, there were outbreaks across the country (Figure 1). By the end of March 2020, there had been more than 81,000 cases and more than 4000 deaths.

Humans play the main role in the transmission of COVID-19. Travel has become more and more frequent and convenient, and geographical distance has become less effective in limiting virus transmission. For example, the city of Wenzhou, which is more than 600 km away from Wuhan, became one of the most seriously affected cities other than Wuhan at the beginning of the pandemic. There are a large number of Wenzhou businessmen engaged in commercial activities in Wuhan, which led to the emergence of imported cases and subsequent local transmission in Wenzhou during the early stage. The nationwide pandemic situation in provinces and cities also reflects this phenomenon. High traffic flow occurs between Wuhan and the dark-colored provinces in Figure 1. The degree of communication between humans is an important factor affecting the spread of the pandemic, in addition to the natural geographical barrier.

The aim of this paper Is to study the transmission characteristics, model, and IPC of COVID-19 in China to help us better understand the transmission and IPC of COVID-19 and hopefully provide other countries with valuable experience and lessons in the prevention and control of this disease.

## 2. Materials and Methods

### 2.1. Routine Reports on the Pandemic

Every day since 21 January 2020, the beginning of the pandemic, the National Health Commission of the People’s Republic of China (NHC) has routinely reported China’s pandemic situation of the previous day. The Health Commission of Hubei Province has been reporting Hubei’s pandemic situation since 31 December 2019. The data reported in this paper start from 20 January 2020 and contain only mainland China, hereinafter referred to as China. During this period, there were several revisions, and we obtained the latest updates. The data sources are the NHC and the Health Commission of Hubei Province. The data used in this study are from the above websites. Figure 2 shows the daily confirmed cases in China as of 31 March 2020. As can be seen in the figure, the first wave of the local outbreak in China basically ended at the end of March. Figure 3 shows the mortality rate of provinces and cities in China as of 31 March 2020.

As a new type of virus, human knowledge of SARS-CoV-2 was limited. At the early stage of the outbreak in Wuhan, people lacked experience and protective measures to manage the virus, and as a result, many infected people without isolation and treatment continued to infect healthy people. A large number of cases occurred in a short time, which put enormous pressure on the healthcare system, leading to issues such as insufficient reagents, testing capacity, and hospital beds. We found that when the healthcare system was overloaded, the mortality rate remained at a higher level. This is because when the healthcare system was overloaded, new cases could not be treated, symptoms worsened, and some patients died. In addition, patients who were already hospitalized could not receive the best available treatment.

### 2.2. Transmission Stages

For the distribution of cases and mortality, we divided cases into those occurring in China, China excluding Hubei Province (abbreviated as China* in Table 1), Hubei Province excluding Wuhan (abbreviated as Hubei Province* in Table 1), and Wuhan.

We divided the community transmission of COVID-19 into the following four stages in China, shown in Table 1:(A)Explosive growth stage: There are new cases every day, and the increase rate is more than 0, i.e., exponential distribution or quasi-exponential distribution.(B)Stable growth stage: There are new cases every day, and the growth rate tends to 0 or fluctuates around 0.(C)Attenuated growth stage: There are new cases or zero growth of less than one transmission cycle every day, and the increase rate is negative, showing negative exponential or quasi-negative exponential distribution.(D)Sporadic growth stage: Zero cases occur in more than one transmission cycle, followed by a continuous period of zero or only sporadic confirmed cases.

### 2.3. Two Special Reports

General testing is mainly based on nucleic acid testing, clinical diagnosis, and epidemiological survey [25,26,27,28,29]. However, it is clearly inefficient when testing takes a long time and the incoming stream of new cases is large. Under these circumstances, untested infected cases can still spread the virus to others. With the in-depth understanding of novel coronavirus pneumonia and the accumulation of diagnostic and treatment experience, and in response to the characteristics of the pandemic in Hubei Province, a new diagnostic method was adopted in Hubei Province to further improve the efficiency of IPC. The new method diagnosed new cases mainly according to clinical signs and epidemiological surveys. This method of diagnosis led to a sharp increase in reported cases 2–3 days after it began on 12 February. This is shown as the abnormally sharp increase in Figure 2a,c,d. In Wuhan, these surges were mainly from the previous backlog and infected individuals who were still in the incubation period without typical symptoms. Cases in other cities in Hubei Province were mainly in the incubation period without typical symptoms. This method actually lowered the diagnostic standard, aiming to find all infected persons and treat them promptly. This is the reason for the first special report.

For the second special report, issued on 17 April 2020, after the pandemic was brought under control, the Health Commission of Hubei carried out a comprehensive survey on the pandemic at the beginning of the outbreak and updated relevant data, mainly for the beginning of the outbreak, resulting in an increase of 325 confirmed cases and an increase of 1290 deaths. At the beginning of the outbreak, the statistics on deaths were omitted, and the reporting channels of confirmed cases were nonstandard and not occurring in real time [30].

In fact, the difference is not only in the mortality rate but also in the distribution of cases. Due to timely and effective control, the total cases in Hubei Province accounted for 83% of all cases in China, the total cases in Wuhan accounted for 61% of the total cases in China, and the total cases in Wuhan accounted for 74% of the total cases in Hubei, as shown in Figure 4. There is a significant difference in the characteristics of virus transmission and prevention among them.

### 2.4. IPC in China

The goals of IPC in China were to break the transmission chain, control and reduce the scale of the COVID-19 epidemic, and treat and provide professional healthcare to all cases to reduce mortality until the end of the pandemic. Approaches involving isolation, such as locking down cities and maintaining social distance, large-scale testing, and isolating and treating all cases, especially in the source and worst areas, could prevent transmission to other areas. These approaches have been implemented since mid and late January [31].

Adjusting measures to local conditions is one of the basic principles of IPC. There are obvious differences between Wuhan and non-Wuhan areas, independent of the number of confirmed cases or the mortality rate. Correspondingly, IPC is much different between Wuhan and non-Wuhan areas.

During the pandemic, it was necessary to wear a mask and maintain social distance when in public. In non-Wuhan areas, isolation and observation for a period of time (usually 14 days) were necessary for personnel returning from epidemic areas, social distance was required, personal protection was encouraged and advocated, and unnecessary personnel movement and gathering were suspended. This kind of IPC generated a transmission pattern of sporadic outbreaks in non-Wuhan areas. In Wuhan, maintaining social distance, stopping unnecessary business activities, limiting the flow of humans, and isolating and treating cases who had been infected in a timely manner were all necessary. Lockdown in Wuhan played a significant role in limiting the increase in new cases and largely restricting cases to local areas without spreading the virus throughout China. This kind of strict IPC generated a band-shaped or sporadic outbreak transmission pattern in Wuhan, rather than a large-scale uncontrollable situation. Generally, in non-Wuhan areas, more specifically, in non-Hubei areas, outbreaks occurred later, the number of cases was small, and the public healthcare system was far below the overload threshold. Compared with other cities in China, the cities (other than Wuhan) in Hubei Province had earlier outbreaks, with a larger number of cases and higher healthcare system utilization. Wuhan was the city with the earliest and largest number of cases, and its healthcare system utilization was the highest in China. With the actual implementation of IPC, the main confirmed cases were limited to Wuhan so as not to transmit the virus to other provinces and cities on a large scale. Medical staff and medical resources from other provinces and cities were sent to Hubei Province, mainly Wuhan [32], to provide support so that Wuhan’s healthcare systems did not collapse.

In terms of testing, IPC in China aimed to find as many cases as possible. In fact, the test capacity was not sufficient to deal with the cases at the beginning of the pandemic. In order to find as many cases as possible, the amount of detection needed to increase. China lowered the test standards for key populations in key areas to find them quickly. All cases should be isolated and treated professionally. Specifically, once the person tested positive, he or she was not able to spread the virus to others. In Wuhan, due to the high number of cases, if these patients were treated in professional hospitals, the healthcare system would have been overwhelmed. Therefore, Wuhan adopted additional effective measures to deal with this problem. With the support of medical resources in other provinces and cities, all infected cases were classified according to their symptoms. All patients with mild symptoms were concentrated in modular hospitals for treatment. Specialized hospitals were used to treat patients with severe symptoms. As of 13 February 2020, there were 14,269 special hospital beds, among which 645 hospital beds were vacant. Moreover, as of 25 February, the number of beds in modular hospitals in Wuhan exceeded 30,000, which far exceeded the number of patients at that time. Modular hospitals play a significant role in improving healthcare system efficiency. On the one hand, all infected persons can be isolated in time to avoid transmission in family clusters [33] or to other contacts, and on the other hand, modular hospitals have professional medical staff to take care of patients and, when patient symptoms become worse, immediately send them to the specialized hospitals. Knowledge of COVID-19 treatment was limited at the time. With the continuous understanding of COVID-19, the treatment plan has continuously improved and been made more effective [25,26,27,28,29,34,35,36,37]. In addition, the Chinese government bears all of the costs of treatment, which has undoubtedly played a vital role in the control of the pandemic.

We used the data from the NHC to model the transmission and to analyze the effect of related IPC at the four stages. Additionally, for the two special reports, we developed a model to estimate and reproduce the daily confirmed cases and deaths.

## 3. Results

We analyzed and modeled transmission in Wuhan and non-Wuhan areas. On this basis, we estimated several parameters of transmission, such as basic reproduction number and incubation period. Furthermore, the daily confirmed cases and deaths were estimated by using the corresponding models.

### 3.1. The Estimation of R_0_

The basic reproduction number $R_{0}$ indicates the average number of people that each infected person infects in a totally susceptible population [38,39]. In the absence of protection, there are A_0_ infected people who can spread the virus to $R_{0}$ people, where $R_{0}\in [0,+\infty )$. After one transmission cycle, the number of new cases is:(1)$$x\left(1\right)=A_{0}\times R_{0}$$

Under these circumstances, after the nth cycle of transmission, the number of infected people is:(2)$$x\left(n\right)=A_{0}\times {\left(1+R_{0}\right)}^{n}$$

We used Equation (2) to estimate $R_{0}$. Then, the province or city data were selected considering three factors as criteria:(a)Data were taken from the beginning of the outbreak.(b)Confirmed cases were sufficient.(c)All cases were tested in a timely manner.

Specifically, for factor (a), at the beginning of the outbreak, people’s awareness of protection was relatively poor, and protection was low, so data during this period will better reflect the ability of the virus to spread. Data meeting the requirement of factor (b) can be better processed and analyzed, and factor (c) means that the healthcare system is operating normally, so there are no omissions, and the data are very likely realistic and reliable.

For the estimation of $R_{0}$ in China, we used the data from Guangdong Province, Jiangsu Province, and the cities of Xiaogan, Huanggang, and Wenzhou. All of these provinces and cities meet the three criteria. In addition, we also used national data in the estimation. The incubation period of COVID-19 is considered to be 4–6 days. Statistics in the literature also cite an incubation period of 5.2 days and 5 days for COVID-19 [40,41]. In order to improve the accuracy of the experimental results, we analyzed the data with incubation periods of 4, 5, and 6 days.

The experimental results show that Wenzhou data have the best fit when the incubation period is $T_{0}=5$ days, with the R-square of 99.59%. At this time, $R_{0}=1.935$ (95% CI). As shown in Figure 5. In the following analyses, we used these values for the incubation period and $R_{0}$.

### 3.2. Study of the Transmission Model in Non-Wuhan Areas

We define the parameter p, which refers to the efficiency of IPC, that is, the measures to break the transmission chain by applying non-medical interventions, including wearing masks, maintaining social distance and home isolation, etc. Here, p∈[0,1].

After the first transmission cycle, there are $\bar{x}\left(1\right)$ infected cases:(3)$$\bar{x}\left(1\right)=A_{0}\times \left[\left(1-p\right)\left(1+R_{0}\right)+p\right]$$

Similarly, after the nth transmission, there are newly infected people, and the number of daily new cases is $\bar{x}\left(n\right)$
(4)$$\bar{x}\left(n\right)=A_{0}\times \left[p+\left(1-p\right)(1+R_{0}\right){]}^{n}$$

The confirmed cases on the same day are cases that have passed the incubation period with the occurrence of clinical symptoms. τ is the incubation period.
(5)$$X\left(T\right)=X^{\prime }\left(T-\tau \right)$$

For IPC in China, firstly, for non-Wuhan areas, that is, areas with sufficient testing capability, we used Equations (4) and (5) to estimate p. We used data from Guangdong Province, Jiangsu Province, and the cities of Wenzhou, Huanggang, Xiaogan, Jingzhou, Huangshi, and Xiangyang. As shown in Figure 6.

The results show that p was greater than 0.62 in some areas when the pandemic situation developed from the stable growth stage to the sporadic growth stage. In other words, when the testing capability was sufficient, essential and appropriate prevention measures (using more than 60% prevention measures compared with not taking any measures) were able to effectively control the pandemic.

### 3.3. Study of the Transmission Model in Wuhan

Wuhan, as the capital of Hubei Province, is an economic center and major transportation hub with a population of more than 11 million people; it is located in central China. China’s current economic structure drives the development mode of cities around Hubei Province, with Wuhan as the core. The transmission also shows a radial pattern from Wuhan to other cities in Hubei Province, as shown in Figure 7 and Figure 8.

In terms of the transmission model in Wuhan, suppose that there are $A_{0}$ infected cases in this scenario. They are found and tested with a probability of t. Once confirmed, they are isolated and treated in a timely manner. Thus, they no longer have the possibility of infecting others. Meanwhile, some cases will not be found and tested with a probability of (1−t). From a statistical point of view, the probability of a false-negative or false-positive test result is very small in China, so it is ignored here. Additionally, we simplified the scenario by assuming that there were no deaths in the whole process. The transmission model in Wuhan is shown in Figure 9.

### 3.4. Revision and Estimate Based on Two Special Reports

Regarding the first special report updating cases on 12 February and 13 February, we assume that the difference is mainly due to the accumulation of the previous period. We do not consider people who died while waiting.

The number of daily confirmed cases is $x_{w}$:(6)$$x_{w}\left(n\right)=A_{0}t\left[p+\left(1-p\right)(1+R_{0}\right){]}^{n-1}$$

The number of daily untested case is $x_{w}^{\prime }$:(7)$$x_{w}^{\prime }\left(n\right)=A_{0}{\left(1-t\right)}^{n}{\left[p+\left(1-p\right)\left(1+R_{0}\right)\right]}^{n}$$

The number of daily new infected cases is $x_{n}$:(8)$$X\left(n\right)=x_{w}\left(n\right)+x_{w}^{\prime }\left(n\right)$$

Given the availability of data, we considered the scenario that all backlogged cases and unreported deaths began to accumulate on 20 January. According to Equations (5)–(8) and Reference [40], incubation periods of 4 days, 3 days, and 2 days were used as well, and we estimated the daily confirmed cases, as shown in Figure 10. According to the NHC report, the total number of reported cases was 32,081 from 20 January to 12 February, and the proposed model estimates that the total number of cases was 32,695 in this period, with an estimated accuracy of 98.12%.

For the study using the second special report, we attempted to estimate and reproduce the death cases at the early pandemic stage. In this part of the process, we used the Poisson process to fit the mortality rate [42,43,44] in Wuhan. According to the information reported by relevant departments, we obtained the regular pandemic reports from 20 January to 12 February, when all cases were recorded, for a total of 24 days.
(9)$$f\left(x|\lambda \right)=\frac{\lambda^{x}}{x!}e^{-\lambda };x=0,1,2,\ldots ,\;\infty$$

We used Equation (9) to calculate the number of deaths on these days, as shown in Figure 11. The total number of reported deaths is 2214, and the total number of estimated deaths is 2207, with an accuracy of 99.5%.

The revised mortality rates of Wuhan during different transmission stages are illustrated in Figure 12.

Among the stages of transmission in Wuhan, the mortality rate was the highest in the explosive growth stage, with a peak value of 13.68% on 27 January 2020. The lowest mortality rate, which was 6.1%, occurred on 11 February during the transitional stage between the stable growth stage and the attenuated growth stage.

### 3.5. Analysis of the Effect of IPC

The revised pandemic transmission stage is consistent with the data shown in Table 1. It can be seen that a series of IPC measures were adopted and implemented in both Wuhan and non-Wuhan areas, contributing to rapid control of the pandemic in China. The IPC measures in Wuhan, including sufficient testing (especially having enough tests to detect all infected people), identifying all cases, and preventing contact with others as necessary, as well as the large-scale implementation of modular hospitals, produced the desired result of reducing the number of infections and deaths from 12 February. According to Reference [45], multipronged IPC had considerable positive effects in controlling the pandemic, reducing the total number of infections in Wuhan by 96.0% as of 8 March 2020.

## 4. Conclusions

Since the large-scale transmission of COVID-19 in January 2020, the virus has infected hundreds of millions of people and caused millions of deaths worldwide, which is a disaster for all of humanity. In this study, we analyzed the transmission model and IPC in China, the country that reported the first outbreak and was the earliest to implement large-scale and effective control of the pandemic.

We retrospectively studied the transmission characteristics, model, and IPC of COVID-19 in China. Due to the lack of knowledge of and experience with COVID-19, including inadequate protection, insufficient testing capacity, and the lack of specialized wards and equipment, especially in Wuhan, Wuhan was significantly affected in the early stage of the outbreak until mid-February. With the accumulation of information on the virus, continuous investment in medical resources, and the implementation of corresponding IPC, the epidemic situation was quickly controlled. In non-Wuhan areas, there was a transmission pattern of sporadic outbreaks, whereas, in Wuhan, the transmission pattern was band-shaped or sporadic.

In addition, we used the proposed model to calculate the reproduction and the incubation period of COVID-19 in China. Furthermore, for two revision reports, we used the proposed model to estimate and reproduce the daily confirmed cases and deaths with high accuracy in Wuhan for the beginning of the pandemic. Based on the revision and estimation, we found that, in Wuhan, at the explosive growth stage, in addition to confirmed cases, deaths also continually increased, and the mortality rate peaked. At the stable growth stage, the confirmed cases remained stable at a high level, and the deaths followed the same pattern, but the mortality rate continued to decline. At the attenuated growth stage, both the confirmed cases and deaths began to decline, leading to a relatively flat mortality rate.

IPC measures, including large-scale testing and the establishment of modular hospitals, effectively contained most cases in Wuhan, avoided large-scale spread in China, and reduced the number of infected cases and deaths. We conclude that with sufficient testing capacity, essential and appropriate IPC can effectively control the pandemic when the situation is not serious or transmission has just started. This view was also verified by subsequent local outbreaks in China. If the pandemic has spread on a large scale, the situation needs to be controlled in a timely manner by adopting strict prevention and control measures, expanding the testing capacity, classifying patients according to their symptoms, and treating them in the healthcare system by using temporary modular hospitals and specialized hospitals. Of course, strict implementation of IPC is the key.

At the moment, the world is still suffering from COVID-19, which is dominated by the Omicron variant. Compared to the reference/ancestral variant, the Omicron variant has a total of 60 mutations, which have made it more infectious [46]. The Omicron variant led to a new pandemic. The symptoms of Omicron are mild. There are 155,150 Omicron cases in Europe, and the severity rate is about 1.36% (as of 20 January 2022) [47]. At the beginning of 2022, the Omicron variant was also reported in the cities of Tianjin and Anyang in China. There were 759 Omicron cases reported in the two cities as of 20 January. There were only four severe cases for a rate of about 0.6%, which shows that the symptoms caused by Omicron itself are relatively mild. However, it is a more infectious virus, and immune escape occurs from time to time. Although vaccination is not 100% effective in preventing infection, it reduces the likelihood of severe illness and death. China still implements strict IPC to control the Omicron variant. A lot of experience has been accumulated in IPC. Vaccines have been verified to be effective against the Omicron variant to some extent, either in preventing infection or in alleviating symptoms. China actively promotes vaccination and has achieved a vaccination rate of more than 90%. In addition, once Omicron cases occur, a multipronged approach that includes large-scale testing, even testing all staff several times, and partial lockdowns has been shown to quickly control the Omicron variant. China adopts a method combining Chinese traditional and Western medicine to treat Omicron cases, and the overall effective rate of traditional Chinese medicine is about 90%.

We maintain that vaccination is a top priority [16,47]. Wearing masks and maintaining social distance are absolutely necessary. Furthermore, large-scale testing is essential. The aim is to find as many infected people as possible. When the test capacity is insufficient, focusing on key groups and mixed testing [48] can be considered. Finally, it must be ensured that there are enough beds for all cases, either professional beds or modular beds, and that all cases receive professional care or guidance so that their state does not worsen. It is hoped that China’s IPC will provide other countries with valuable experience and lessons in the prevention and control of COVID-19 and that, in the future, humanity will face new challenges calmly and rapidly with a well-thought-out plan for new infectious diseases.

## Figures and Tables

**Figure 1 ijerph-19-03074-f001:**
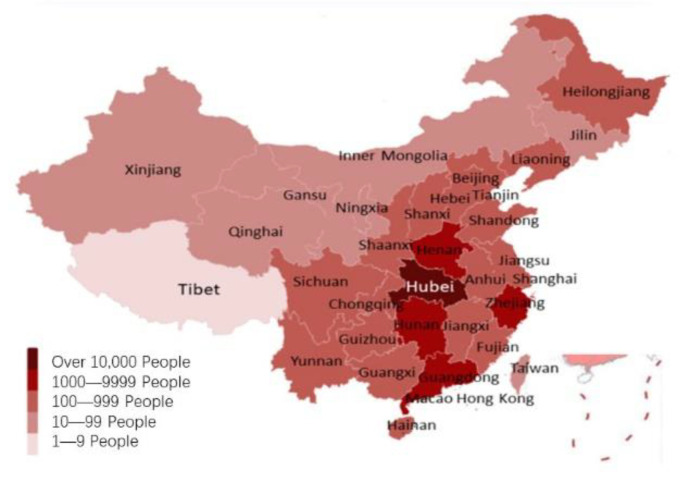
The pandemic situation in China on 31 March 2020.

**Figure 2 ijerph-19-03074-f002:**
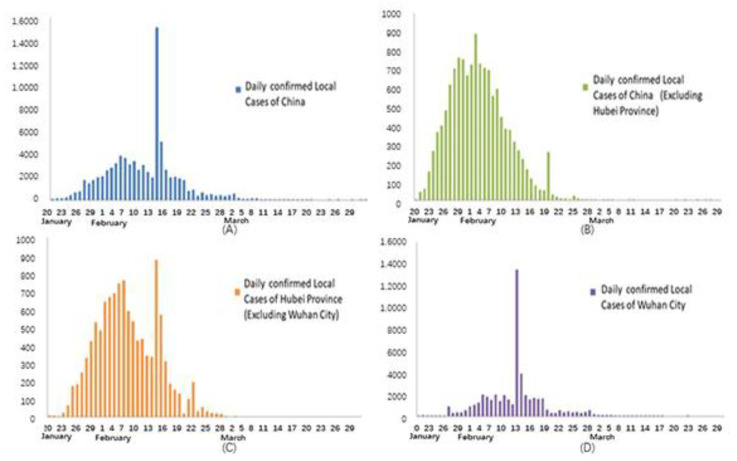
Daily confirmed local cases in China and the most affected areas, shown in (**A**–**D**). The figures show that, in the most affected areas in China, including Wuhan and Hubei Province, the first wave of the COVID-19 pandemic basically ended in the middle and late March and entered the sporadic stage. The pandemic in the less affected areas of China ended earlier.

**Figure 3 ijerph-19-03074-f003:**
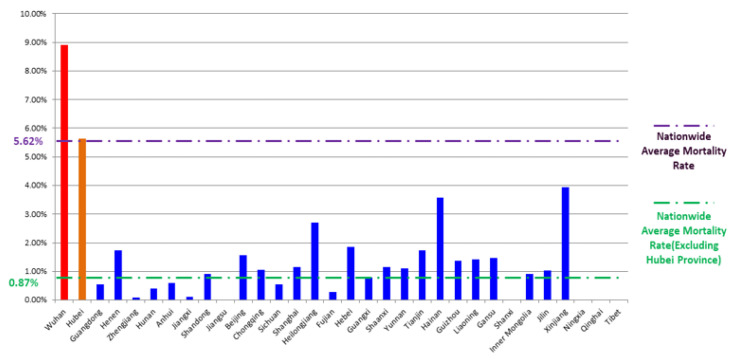
The mortality rate in Wuhan and all provinces and cities. Nationwide, as of 31 March, the mortality rate in Hubei Province, especially Wuhan, was higher than that in non-Hubei areas. The mortality rate in non-Hubei areas was only 0.087%, while the mortality rate was 5.64% in Hubei Province, 3.57% in the non-Wuhan area, and 8.91% in Wuhan. The national mortality rate was 5.62%.

**Figure 4 ijerph-19-03074-f004:**
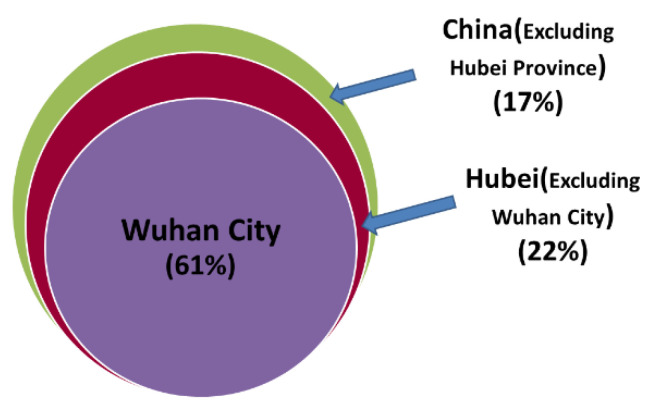
The distribution of confirmed local cases. Wuhan accounts for 61% of all cases in mainland China; Hubei Province accounts for 83% of all cases in mainland China, excluding Hubei Province; and cases in other provinces and cities accounted for 17% of the total in mainland China. Hubei Province, as the area most affected by the pandemic, accounts for the majority of the total cases in China, compared with the total population size of other provinces and cities.

**Figure 5 ijerph-19-03074-f005:**
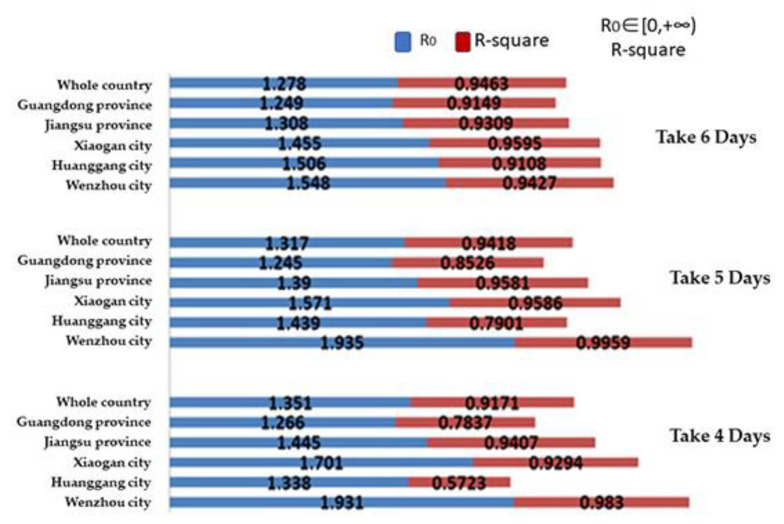
The evaluation results of $R_{0}$ and the corresponding R-square.

**Figure 6 ijerph-19-03074-f006:**
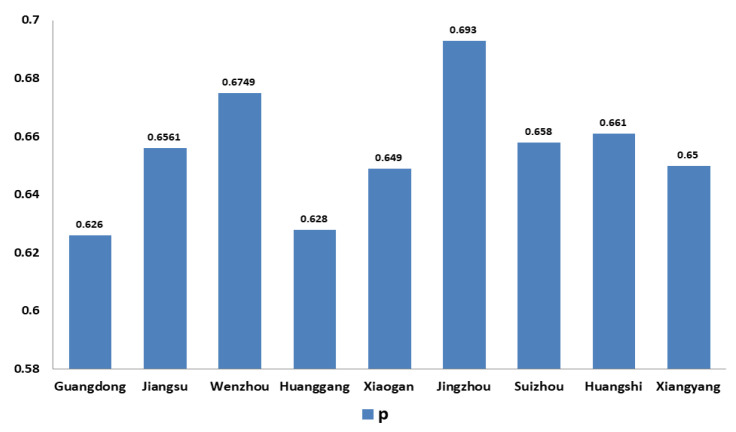
*P*-values of several cities (95% CI).

**Figure 7 ijerph-19-03074-f007:**
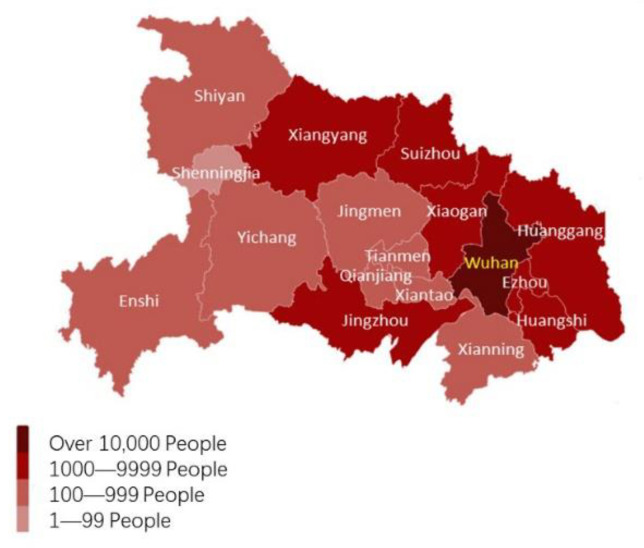
Pandemic situation in Hubei Province.

**Figure 8 ijerph-19-03074-f008:**
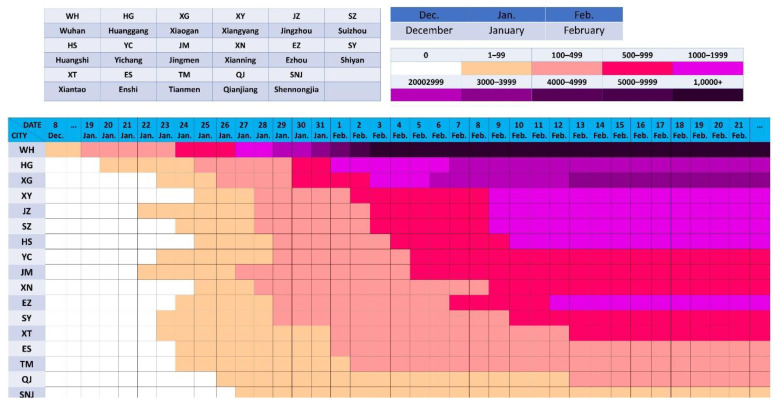
The time of the first reported cases and the following transmission situation in cities and prefectures in Hubei Province. In Hubei, there are cases in all cities and prefectures, and the order of appearance is related not only to the geographical location of Wuhan but also to the degree of connection.

**Figure 9 ijerph-19-03074-f009:**
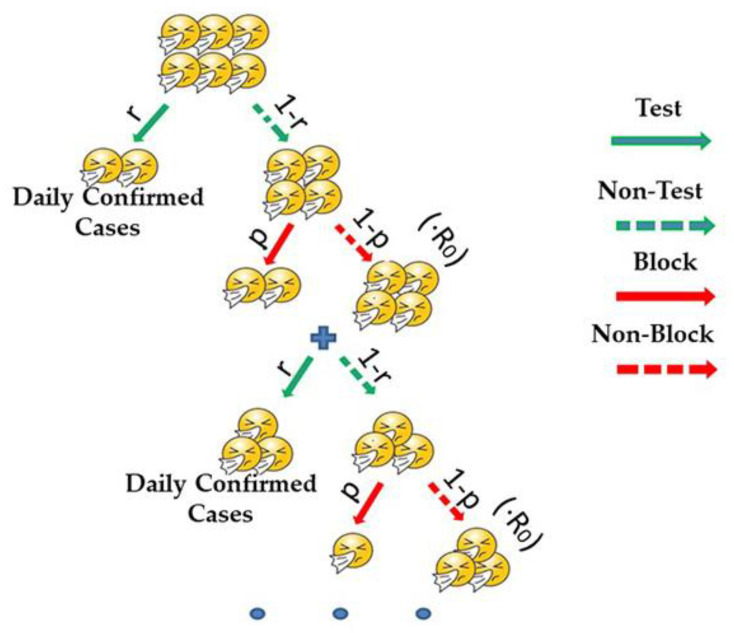
The transmission model of COVID-19 in Wuhan. Here, t represents the probability that an infected person will be tested. For example, if there are 100 infected people, but because of the limited test capacity, only 1000 people can be tested in one day, and 960 people tested are not infected, then only 40 infected people are tested; thus, at this time, t=0.4. Therefore, there are 60 unconfirmed infected people at risk of infecting others with a probability of p.

**Figure 10 ijerph-19-03074-f010:**
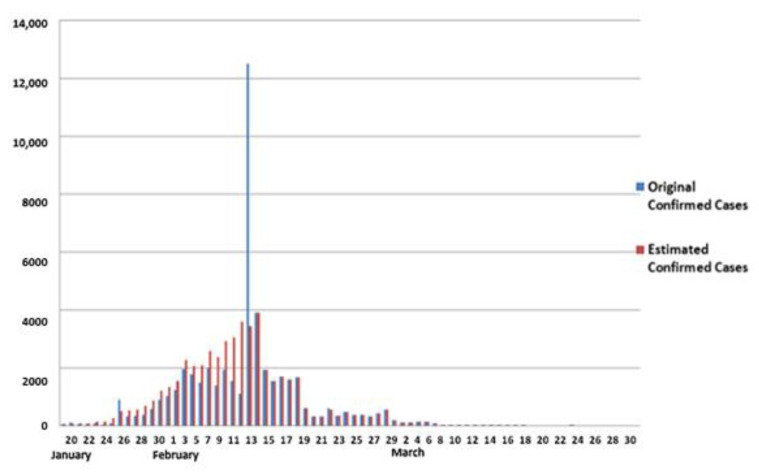
Original confirmed cases and estimated confirmed cases in Wuhan.

**Figure 11 ijerph-19-03074-f011:**
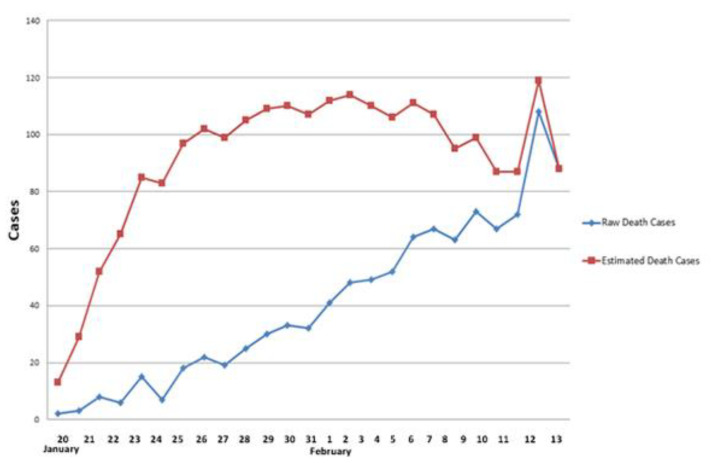
Original and estimated death numbers in Wuhan from 20 January to 13 February 2020.

**Figure 12 ijerph-19-03074-f012:**
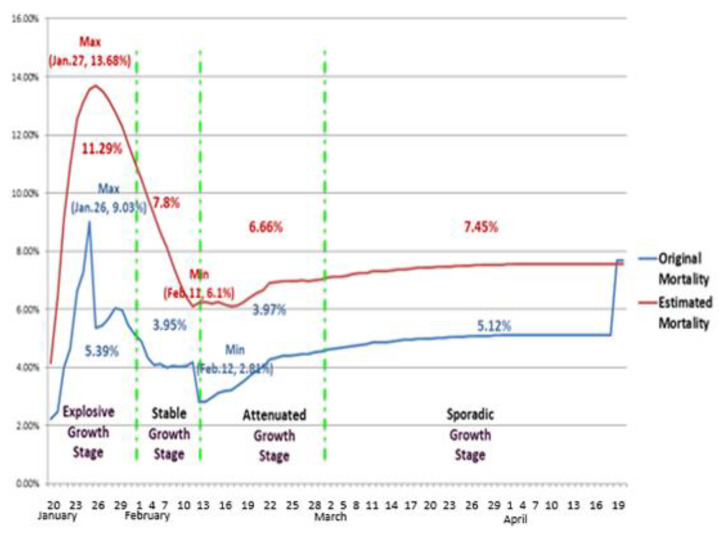
Original mortality rate and estimated mortality rate in Wuhan. The blue line represents the original mortality rate, and the red line represents the estimated mortality rate. Our results show that the average mortality rate is the highest in the explosive growth stage, 7.8% in the stable growth stage, and 6.66% in the attenuated growth stage. In the sporadic growth stage, the average mortality rate is 7.45%.

**Table 1 ijerph-19-03074-t001:** Four stages of transmission of COVID-19 in China.

Stage Area	Explosive Growth Stage	Stable Growth Stage	Attenuated Growth Stage	Sporadic Growth Stage
China	December 2019–4 February 2020	5 February–18 February 2020	19 February 2020– Present	Unknown
China (excluding Hubei Province)	20 January–29 January 2020	30 January–8 February 2020	9 February–6 March 2020	7 March 2020–Present
Hubei Province (excluding Wuhan)	21 January–4 February 2020	5 February–18 February 2020	19 February–3 March 2020	4 March 2020–Present
Wuhan	December 2019–2 February 2020	3 February–13 February 2020	14 February–17 March 2020	18 March 2020–Present

## Data Availability

Not applicable.

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
