# Peer review of "The Transmission, Infection Prevention, and Control during the COVID-19 Pandemic in China: A Retrospective Study"

_ijerph, 2022, doi:10.3390/ijerph19053074_

Round 1
Reviewer 1 Report
Dear Authors,
I read your publication with great interest.
The data studied in your paper, provided by Routine and next by special reports of pandemic gave a useful conclusion.
Four stages of transmission of Covid-19 in China were clearly defined and descriptive.
Your Study on the transmission model in Wuhan and non-Wuhan areas was conclusive.
The accuracy of the proposed model (98.12% and 99.5%) was high and I hope it can be used in the future.
Author Response
Comment 1: The data studied in your paper, provided by Routine and next by special reports of pandemic gave a useful conclusion. Four stages of transmission of Covid-19 in China were clearly defined and descriptive. Your Study on the transmission model in Wuhan and non-Wuhan areas was conclusive. The accuracy of the proposed model (98.12% and 99.5%) was high and I hope it can be used in the future.
Response 1: Thank you for your comments. In this paper, we have retrospectively studied on the transmission, infection prevention, and control (IPC) under the COVID-19 pandemic in China. We studied the transmission characteristic of COVID-19 in China and IPC. Furthermore, we proposed models to estimate and replay the daily cases and death cases for the beginning of the pandemic.
It is no doubt that COVID-19 is a disaster for all mankind. We hope the world will recover health as soon as possible. Finally, we hope you and your family are healthy and safe!
Reviewer 2 Report
1 In the introduction, the objective of the study is not stated.
2 In materials and methods, it does not describe the methodology used, what type of study they carried out.
3. In results, they describe information that corresponds to the methods section.
Author Response
Comment 1: In the introduction, the objective of the study is not stated.
Response 1: We would like to express our sincere appreciation for your careful reading and invaluable comments. We have studied the comments carefully and have made corrections which we hope to meet with approval. China has a relatively complete pandemic transmission process and effective infection prevention, and control. We hope that through the retrospective study of the pandemic, we can understand COVID-19 well, and provide other countries with valuable experience and lessons. We have added the objective of the study on page 2, and in the manuscript are highlighted in yellow color.
Comment 2: In materials and methods, it does not describe the methodology used, what type of study they carried out.
Response 2: According to the data of the National Health Commission, we studied the characteristics of transmission, including analysis and modeling, and proposed transmission models. We have added the methodology used on page 6, and in the manuscript are highlighted in yellow color.
Comment 3: In results, they describe information that corresponds to the methods section.
Response 3: We have added the corresponding description in results section on page 6, and in the manuscript are highlighted in yellow color.

Reviewer 3 Report
Dear Editor and authors,
Thank you very much for this opportunity to review this timely important article. This is an interesting article from China concerning the transmission, infection prevention, and control of the Covid-19 pandemic in China. However, the interest at national level, the information included in this article is of limited international value. In addition, there are now some information missed in the submission that could be of interest to the health authorities in order to implement adequate measures.
Please include information below as you can as Omicron variant is now the global concern for healthcare authorities. I believe the inclusion of the below information would befits to grab international interest to their article.
1. Change Covid-19 into COVID-19 in the whole text body.
2. How about the effective prevention and therapeutic managements against Omicron? How about the clinical phenomes and therapies? How about the clinical and translational medicine?
3. Whether the COVID-19 vaccinations still function in preventing from Omicron variants and whether life-saving protection and vaccinated populations can be re-infected? What is the impact on cases, hospitalizations and deaths that is provided by vaccination and by boosters?
4. What are the Omicron variants numbers and sequencing of mutations?
Author Response
Thank you very much for this opportunity to review this timely important article. This is an interesting article from China concerning the transmission, infection prevention, and control of the Covid-19 pandemic in China. However, the interest at national level, the information included in this article is of limited international value. In addition, there are now some information missed in the submission that could be of interest to the health authorities in order to implement adequate measures.
Please include information below as you can as Omicron variant is now the global concern for healthcare authorities. I believe the inclusion of the below information would befits to grab international interest to their article.
Comment 1: Change Covid-19 into COVID-19 in the whole text body.
Response 1: Thank you for pointing it out. We are sorry for this problem and have corrected it according to your suggestion. We have changed all “Covid-19” into “COVID-19” in the whole text body in turquoise color.
Comment 2: How about the effective prevention and therapeutic managements against Omicron? How about the clinical phenomes and therapies? How about the clinical and translational medicine?
Whether the COVID-19 vaccinations still function in preventing from Omicron variants and whether life-saving protection and vaccinated populations can be re-infected? What is the impact on cases, hospitalizations and deaths that is provided by vaccination and by boosters?
What are the Omicron variants numbers and sequencing of mutations?
Response 2: Thank you very much for the significant suggestion. We are concern about the Omicron variant as the new global concern for healthcare authorities. According to your suggestion, we have studied the latest research about Omicron, and added relevant contents in combination with the prevention and control of Omicron cases in China. The relevant contents are highlighted on pages 1, 12, 13 of the manuscript in turquoise color. At the same time, 3 references have been added on pages 1 & 12 (ref 16, ref 46, ref 47).

Reviewer 4 Report
This paper is an interesting retrospective study sobre la transmisión, prevención y control de la infección durante la pandemia de Covid-19 en China. The paper is well written, easy to read and well referenced and up to date.
My main criticism is that the authors state that strong and effective IPC with strict implementation can effectively and quickly control the pandemic but they do not detail what the IPC specifically consists of. Detailing this information is important for countries where covid-19 prevention and control measures have been lax.
Author Response
Comment 4: My main criticism is that the authors state that strong and effective IPC with strict implementation can effectively and quickly control the pandemic but they do not detail what the IPC specifically consists of. Detailing this information is important for countries where covid-19 prevention and control measures have been lax.
Response 4: Thank you for pointing it out. We are sorry for your confusion because we didn't express it clearly. We have carefully checked the manuscript. In Section 2.4, we introduce the approaches of IPC on page 5 in blue words, followed by the corresponding description on pages 5-6. In order to state more clearly, according to your suggestion, we have added some detailed information about IPC on pages 6 & 13 and in the manuscript are highlighted in green color. At the same time, 3 references have been added on page 13 (ref 16, ref 46, ref 47).

Round 2
Reviewer 2 Report
I want to thank the authors for accepting the suggestions that from my point of view enriched the article.